# Profile of Dogs’ Breeders and Their Considerations on Female Reproduction, Maternal Care and the Peripartum Stress—An International Survey

**DOI:** 10.3390/ani11082372

**Published:** 2021-08-11

**Authors:** Natalia Ribeiro Santos, Alexandra Beck, Cindy Maenhoudt, Charlotte Billy, Alain Fontbonne

**Affiliations:** 1Unité de Médecine de l’Elevage et du Sport (UMES), Ecole Nationale Vétérinaire d’Alfort, Maisons-Alfort, 94700 Paris, France; cindy.maenhoudt@vet-alfort.fr (C.M.); alain.fontbonne@vet-alfort.fr (A.F.); 2Ceva Santé Animale, 33500 Libourne, France; alexandra.beck@ceva.com (A.B.); charlotte.billy@ceva.com (C.B.)

**Keywords:** dog breeding, international, survey, maternal care, behaviour, peripartum, stress

## Abstract

**Simple Summary:**

Nowadays, all over the occidental world, an increasing number of dogs are bred to be companion animals and they are often considered part of the family. Although, dog overpopulation is still a current problem, the importance of dogs as companion or work animals is undeniable. The source of the dog (size and type of kennels) seems to affect its performance and its life outcome. To understand the perception of dog breeders and puppy production in western societies, an online international survey was sent to breeders in North and South America, Europe and Oceania. The survey focused on breeders’ demographics and maternal behaviour and stress during the peripartum period. Overall, based on the responders’ feedback, puppy production is an activity driven partially by small-scale hobby or sporadic breeders. A close relation between dogs and breeders has been observed in different countries with small differences. In general, stress reduction during strategic times was a common concern amongst breeders who were using slightly different interventions. Spending more time with the bitches to reassure them was commonly employed.

**Abstract:**

Dogs are part of many households worldwide and in recent years in some western countries, more and more people describe them as members of the family. The accurate number of dogs per household and the actual demand for puppies are hard to estimate correctly. The sources of dogs seem to be an important element to consider before acquiring a dog since some behavioural disorders are more likely observed in dogs coming from puppy mills and pet stores. Therefore, there is an increased need to better-know dog breeders, theirs believes and practices. To full-fill this objective, an online questionnaire in five different languages (English, German, Polish, Portuguese and Spanish) was sent to dog breeders. The survey intended to study the demographics of dog breeders and to understand their perception and practices about reproduction, maternal behaviour and management of the dam during the peripartum period. We were also interested to evaluate the occurrence and the impact of stress around parturition and how breeders deal with overstressed bitches and if they believe that motherly quality could have a later-on effect on the livelihood of the dog. Six hundred and sixty-eight respondents participated in the survey, mainly from Australia, Brazil, Canada, Germany, Portugal, Poland, Spain, United Kingdom, United States of America and non-listed country (reported as ‘other’ in the survey). A country effect was observed in relation to housing system, human interaction, the use of techniques to define breeding and whelping time and different methods to manage stress during the peripartum period. Considering the demographics of the responders, breeding activity is, in general, a family based activity with an effect of country. Use of progesterone levels to manage reproduction and pregnancy diagnosis by ultrasound were cited as common practice. In general, parturition takes place under human supervision. Peripartum period was perceived as a stressful moment with a difference in the way of addressing the problem based on the country but reassuring the bitch by increasing human presence was frequently described. Inappropriate maternal behaviour was observed by half of responders and if presented primiparous bitches and parturition by C-section were overrepresented. Puppies stayed with the dams from 4 weeks to 9 weeks and dog breeders from Spain, Poland and Portugal are keeping puppies with their mother the longest. Overall, poor maternal behaviour has an impact on puppies’ cognitive development with German breeders being more convinced than their counterparts from Brazil and Spain.

## 1. Introduction

The human collaboration with dogs plays a very special role in the foundation of society in many countries. Several factors led to domestication and socialisation of dogs, making this interspecies collaboration one of the oldest animal–human interactions [1,2]. This partnership not only shaped civilisation [2,3,4] but also had an impact on the strategies of adaptation of dogs [5,6,7]. Many studies have tried to clarify the reasons to explain this interaction. The more recent hypothesis is a convergent evolution between dogs and humans [2] leading to genetic predisposition to improve the skills of cooperation and communication [4,5,8,9,10,11]. The benefits of the human–dog interaction [12,13,14,15] rank dogs as one of the most beloved pets, especially in western cultures [16]. However, the reasons associated to a breed choice and motivation of breeders for reproductive decisions has received little contemplation.

Despite dog overpopulation being still a major problem worldwide, in some western societies there is an increasing number of people that consider dogs as family members [17]. Consequently, there is a constant need to address global dog population control, but also, it is important to know the origin of dogs (profile of breeders, reasons behind breeding decisions and socialisation procedures used) for a better care for pets. It is hard to estimate the dog population by country and therefore in the world. Considering the number of dogs per household, the USA seems to have the highest dog population, followed by Brazil, China and Russia [18,19]. The dog population worldwide is estimated to be 470 million dogs kept as pets [20] and a global population of over 900 million [21].

Globally, the market of companion dogs is growing, leading to a public concern about welfare. In particular, one question raised is how to meet the demand and at the same time guarantee a sustainably and ethical source of companion dogs [22]. A recent publication discussed the shortage of puppies in the UK due to the increasing demand related to the lockdown and homeworking during the COVID-19 pandemic [23]. All these aspects have led to more interest from researchers into the dog breeding industry [24,25,26]. Recently, there was also an expanding initiative, in various countries, to better regulate or tighten existing regulations on standards of care for breeding dogs.

For a prospective dog owner there are many options for obtaining their pet including breeders, ads on the Internet, pet shops, rescue organisations and shelters. The decision of what dog to acquire is multifactorial [27] taking into consideration size, breed, age, coat colour, health and purebred status [28,29,30]. Appearance also seems to be important [31,32,33] as well as behaviour and temperament [31,32,34]. Clearly, breeding kennels are highly linked to the looks and the temperament of their dogs. Overall, dog breeders can be divided into categories non-exclusively defined as purebred, familial, occasional and commercial breeders with the activity being the primary or secondary income. The classification is based on the number of animals dedicated to reproduction, the number of litters per year, the type of housing for the breeding stock and the impact of financial gain on the outcome of the breeder. In the USA, the number of commercial breeding enterprises has increased [22]. While it seems that a large proportion of litters are born in unregulated conditions, the public is pushing for more regulation of dog-breeding facilities [35]. In Australia, those regulations vary according to the category of breeder; some commercial breeders, known as puppy farms, and backyard/occasional breeders are covered by little or no regulatory framework [26]. In Europe, despite specificities according to each country, all dog breeders need to be registered under a new European Union Animal Health Law and comply with the Responsible Dog Breeding Guidelines since 2020 [36]. In Brazil, the importance of dogs and cats as pets, as well as the demand for high standard veterinary services, has increased over the past decades [37]. All the efforts in different countries aim at better controlling puppy production and welfare of breeding animals.

Undeniably, it is important to know the origin of the puppy and the characteristics of the breeding facility since the impact of early experiences might have a long-lasting effect on the character of the dog in adulthood. When selecting a suitable specific breed, the parents’ performance and the breeding structure are not the only considerations, as the maternal behaviour and specific events around weaning time and during socialisation can also influence the puppy behaviour. Moreover, the source of the puppies seems to be correlated with the prevalence of behavioural problems [38]. According to these authors, dogs coming from puppy farms and pet stores are more likely to develop behavioural disorders later in life as fear and aggression [38,39]. This has a direct impact on dogs’ welfare since behavioural disorders are considered a primary cause of dog abandonment [22,40,41,42].

Better knowing the motivations and breeding practices could improve the interaction between veterinarian and people seeking a dog, to guide them to breeders that are highly motivated to produce healthy animals. In a recent survey done in the USA, one complaint from breeders was the lack of knowledge of veterinarians in their activity [43]. Understanding breeder’s objectives and practices could improve the trust amongst different players in the sector and ultimately improve dog welfare. The objective of this survey was to collect data on dog breeders from different countries, their practices in relation to the reproduction and identify their perception and management of maternal behaviour and potential peripartum stress in the bitches.

## 2. Methods

### 2.1. Web-Based Survey

Data were collected by an online survey and participants were recruited via the Internet, including breeder websites and breeder social media channels such as Facebook and WhatsApp. In addition, veterinarians specialised in reproduction (members of the European Veterinary Society for Small Animal Reproduction located in all of the selected countries) were contacted to help diffuse the survey in different countries. To increase the participation, the investigators also reached out to individual breeders as well as several Kennel Clubs and breed associations from different countries and asked them to help propagate the questionnaire.

The questionnaire, an updated version of a survey conducted in France in 2019 [25], was available from 1 June 2020 until 10 October 2020 and consisted of five sections. An introduction explained the purpose of the research. No treatments or interventions in the life of breeders and their dogs were proposed. The questionnaire was available in five different languages (English, German, Polish, Portuguese and Spanish). To access the survey use the link for each country. https://drive.google.com/drive/folders/1qgCPW5fFBdAZs8gdrVjEnmRNHacWEX88?usp=sharing (accessed on 4 June 2021). The languages were chosen based on the planned strategies to disseminate the survey as well as to reflect the pet dog market as requested by the project’s sponsor. Respondents chose to participate freely via the Internet.

The structure of the survey is summarised (Table 1) and it was used to formulate the different questions. Overall, this survey consisted of 60 mixed questions (multiple choice and open) divided in five continuous sections that could be completed in approximately 15 min. Responses to all questions were required to submit the survey (no blank responses were allowed). Finally, to encourage more breeders to answer the questionnaire, identification was not required, although breeders could add an email address in the first part of the survey.

### 2.2. Statistical Analysis

Parameters were described overall and according to country. For each parameter, frequency counts and percentages were provided. For relevant parameters, such as frequency of daily human interaction and methods to estimate the breeding time, to estimate the whelping time, to manage stress before/during whelping and after parturition and length of stay of puppies with mother, a logistic regression was performed. The number of bitches (≤5 or >5), the type of activity (main source of income, secondary source, sporadic activity) and the country were used as covariates. For parameters relative to the method to manage the stress before/during/after whelping, frequency of daily human interaction was also used as a covariate. Results of these models allowed to evaluate if the distribution of the parameter was different between breeders among different countries (or number of bitches or type of sporadic activity) or not. For modelling, all relevant parameters were classified in two categories:All the time or almost/half of the working day (around 6 h—morning or afternoon) or less for the frequency of daily human interaction;Yes/No for the use of each specific method;Eight weeks or less/9 weeks or more for the length of stay of puppies with mother.

All models were performed on an exploratory way, so type one error (α) was set at 5%. SAS Software (Version 9.4) was used to provide the statistical results.

### 2.3. Ethics Statement

In accordance with the General Data Protection Regulation (GDPR), the respondents were free to choose whether to participate in the survey and the data obtained only addressed the stated objectives of the research. A statement was provided in the survey introduction to assure that responses were confidential, and the information collected would be used for research purposes only. In accordance with the regulations on personal data, any respondent has rights of access and rectification and limitation of processing of their data. They may also oppose the processing, withdraw their consent, request the deletion or portability of their data by completing a form available from Ceva (https://www.ceva.com/en/Footer-s-links/Privacy-policy3#contact (accessed on 4 June 2021). Portability is the common term used in GDPR. The data subject has the right to receive the personal data concerning him or her, which he or she has provided to a controller, in a structured, commonly used and machine-readable format and has the right to transmit those data to another controller. In addition, the contact information of one of the investigators was provided, as well as the name of the institution behind the survey.

No financial or gift compensation was proposed for completing the questionnaire. It was simple to answer and did not ask for any interference with the daily life of breeders and their kennels. Based on the methodology applied no review by an ethics committee was necessary.

## 3. Results

### 3.1. Breeder’s Profile and Business Organisation

Six hundred and sixty-eight respondents participated in the survey. The distribution based on the country is summarised (Figure 1) and pooled by geographical region (Figure 2). Since the number of Canadian breeders who responded to the questionnaire was small (*n* = 10), despite all the efforts to contact them, their answers were combined with the USA. Almost half of the responders were European breeders (310/668). The ‘other category’ describes responders whose country was not listed in the proposed options. To submit the questionnaire all questions needed to be completed, therefore all responses were used in the statistical analyses. Overall, breeding activity was considered as a hobby or sporadic activity for more than 90% of respondents, with Brazil and Spain as two exceptions, where 10/76 (13.2%) and 10/66 (15.2%) defined their activity as the main source of income, respectively. The profiles of the kennels, in terms of size of the kennels (defined by the number of breeding bitches) were in accordance with the classification of the activity. The vast majority of breeders (624/668) had less than five mating bitches, except breeders from Brazil and Spain where 22.3% (17/76) and 19.7% (13/66) have more than ten reproducing bitches. Comparatively, only 0.76% (4/526) of breeders produced more than 11 litters per year from all other countries. Most breeders (79.5%) only raised one breed (531/668) and the group 8 (Retrievers, Flushing dogs and Water dogs) was the most represented (157/668), based on the classification of the *Fédération Cynologique Internationale* (www.fci.be (accessed on 4 June 2021). Then, groups 2 (Pinschers and Schnauzers—Molossoids, Swiss Mountain and Cattle dogs), 1 (Sheepdogs and Cattle dogs) and 9 (Companion and Toy Dogs) respectively followed it closely (Figure 3).

### 3.2. Housing System and Human Interaction

The housing system when the breeding bitches were not pregnant was very heterogeneous amongst countries. Housing at home exclusively or with access to a garden were the most common responses, but breeders in Brazil, Portugal and Spain used a kennel more frequently than elsewhere (Figure 4).

Despite the housing system, close human contact (Figure 5) was a common point with a small sample 3.1% (21/668) of responders saying the contact was only during two periods per day and 0.4% (3/668) only once a day. More than 80% of breeders from Germany, Poland, USA/Canada and United Kingdom answered that they interacted with the dogs all the time. Germany was the place where breeders declared the highest level of human interaction (91.8% (56/61)). In contrast, more than 30% of breeders from Portugal and almost 30% from Spain selected the options half of the working (during the morning or during the afternoon) day or twice a day. In addition, the size of the kennels affected the levels of interaction. The percentage of breeders with five bitches or less providing their bitches with daily human interaction all the time or almost was higher than those breeders with more than five bitches (*p* = 0.014).

Transferring the bitches to a specific whelping area around parturition time was not a common practice amongst different countries. It seems more routinely done in Brazil, Portugal and Spain. For breeders from USA/Canada, Germany, United Kingdom and the other combined countries, the introduction of a whelping box without moving the bitch from its habitat prevailed (Figure 6).

### 3.3. Usual Practices Used around Breeding, Pregnancy and Whelping

Although responders cited diverse techniques to optimise reproduction, the most common method (55.2%) was the use of progesterone blood levels (P4) followed by observation of the bitch’s behaviour during oestrus (20.8%) (Figure 7) with similarities amongst different countries (Figure 8). Progesterone assays were commonly used with the exception of Brazil, where only 23/76 (30.3%) of responders made use of progesterone levels to estimate the breeding time as the main method, compensated by the use of vaginal smears. Behaviour of the bitch was the second most often method, whatever the country, used to decide the breeding time except in Brazil where evaluation of vaginal smears was placed in second. The size of the kennel influenced the use of different methods to estimate the breeding time. Progesterone assays to estimate the breeding time was done significantly more frequently for breeding kennels with five or less bitches than those with more than five bitches (*p* = 0.008). Conversely, the use of a teaser male was significantly more implemented in kennels with more than five bitches than those with five bitches or less (*p* = 0.046).

In relation to the question about pregnancy confirmation only 12.4% (83/668) of responders said this procedure was not done. If pregnancy diagnosis was performed, ultrasound was the common method in over 80% of the responders irrespective of the country (Figure 9), with a significant effect of kennel size in favour of dog breeders with ≤5 bitches compared to dog breeders with more than five bitches (*p* = 0.040).

If some common methods were observed for estimation of breeding time and pregnancy diagnosis, the procedures to estimate the time of parturition were globally more diverse across the countries (Figure 10). However, regardless of countries, breeders applied in similar proportion (around 20%) the calculation from the date of breeding or ovulation, observation of behaviour and the drop-in the rectal temperature around whelping time to predict the day of parturition.

Breeders in USA/Canada and Australia rely more on progesterone levels as an indicator of ovulation to calculate the whelping day, with a significant difference (*p* = 0.0054) between USA/Canada and other countries as well as between responders from Australia and other countries. For European breeders, different methods were applied in a similar way (around 20%). In Brazil, the behaviour changes of the bitch around parturition was an important clue for breeders, as opposed to the USA/Canada where it is poorly used (Figure 11). Behaviour changes observed included refusal to eat and appetite reduction, restlessness, circling, scratching the floor (attempting to make a nest), etc.

Calculation from the day of ovulation was used significantly less often in Brazil and Spain compared to all other studied countries (except for Spain/Portugal comparison: *p* = 0.0578). These countries rely much more on behaviour observation, and the opposite distribution was declared in USA/Canada.

### 3.4. Management, Signs of Stress and Behaviour of the Bitch in the Peripartum Period

The majority of breeders (71.3%) managed to change the housing for the maternity period (448/668). Half of them (244/488) transferred the bitch to a specific area (maternity) before parturition while the other half introduced a whelping box where the bitch was housed (202/488) or changed the housing in another way (41/488). However, disparities were observed according to the country. Brazilian, Portuguese and Spanish breeders transferred their bitches to a separate area (respectively 71.1%, 65.9% and 65.6%) much more than breeders in USA/Canada and Germany (26.9% and 21.1%) who tended to prefer to introduce a whelping box.

Whelping was under close surveillance irrespective of the country. Staying close to the bitch at parturition was the commonest method, accounting for at least 95% of the cases, except for Brazil and Spain where they also combine this with regular visits to the bitch (Figure 12).

#### 3.4.1. Signs of Stress around Parturition

A common concern amongst breeders from different countries was to reduce the stress of bitches around parturition. Signs of stress most commonly described by the breeders were agitation, pacing and restlessness (38% of all 668 breeders), followed by aggressiveness towards people (34.2%), barking and whining (32.3%), panting (31.1%) and trembling (28.4%), although some of those signs can also be part of the normal physiological manifestations of parturition initiation. According to breeders in the survey, the intervention to reduce stress is a preventive or curative action (Figure 13). Less than 10% of the breeders, all countries pooled together, declared they would not interfere to reduce stress. Similarly, 16.8% of the breeders (112/668) selected none of the proposed possible signs of stress, and declared that their bitches never showed signs of stress before whelping.

For breeders addressing stress management (*n* = 618), the chosen method to reduce the stress of the bitch was direct interaction with the dam around parturition time in more than 70% of the cases (Figure 14), either by spending more time with the bitch or providing massage to them. The proportion of dog breeders spending more time with the bitch to reduce stress before/during whelping was not statistically different according to country, type of activity, number of bitches and frequency of daily human interactions.

The way to manage stress revealed a country effect for several types of interventions including playing music using a device such as a radio in the maternity area, and using natural products.

Based on percentages, dog breeders from Portugal played significantly more music in the maternity area to reduce stress before/during whelping than dog breeders from Brazil, Germany, Poland, Spain and other countries (Figure 15). Interestingly, for this method (playing music), a statistical difference was associated with the breeders’ activity. Breeders who described the activity as their main source of income played music as a method to de-stress bitches more often than those labelled as a second source of income (*p* = 0.0209) and sporadic activity (*p* = 0.0008). Breeders that labelled their activity as a secondary source of income played music more frequently than those that labelled their activity as a sporadic activity to reduce stress (*p* = 0.01).

The use of natural products (pheromones, Bach flowers, oral calming supplements, homeopathy, phytotherapy, essential oils.) was quoted by less than 15% of responders (Figure 15) from each country (except in Germany, with 23.6%). For those using natural products to reduce stress (*n* = 65) oral calming supplements and phytotherapy were the less often considered options.

#### 3.4.2. Signs of Stress after Parturition

In general, bitches were allowed to leave the whelping area; only 5% (34/668) said the access was limited to less than three times/day. For the majority (82.4%), the access outside was freely or on demand with a country effect. Spanish and Brazilian breeders were more likely to allow bitches to go out at specific times (respectively 68.2% and 69.8%).

In relation to maternal behaviour, over 80% of all breeders agreed that bitches display a motherly attitude soon after parturition. The quality of maternal behaviour was measured by close interaction of the dam with her puppies, frequent licking and nursing. Conversely, stress-induced inappropriate behaviour was associated to refusing interaction with the offspring (86.1%), refusing to nurse (77.2%), aggressiveness towards puppies (68.7%) and frequent displacement of puppies (64.4%). Comparing only the breeds that were predominantly represented (*n* = 205 responders), there was no statistical difference among countries (*p* = 0326) and breeds (0.066) concerning the time to display maternal behaviour (normal—right after puppies’ birth—or abnormal). The stress after parturition was a common worry amid breeders, with more than fifty percent taking preventive actions to address the problem (Figure 16).

The way to deal with stressed bitches after parturition was similar to the stress management around whelping time, as described before. Once again, reassuring the bitch by spending more time with her was the most common method (86.5% on average). Playing music was also an option for 7.8% of breeders from Germany and up to over 30% for breeders in USA/Canada and Portugal. For breeders from Brazil, Germany and Poland, playing music as a method to reduce stress was significantly less exploited than those from United Kingdom, Australia, Portugal and USA/Canada. An effect of breeders’ classification of their activity was observed for the use of music after parturition (*p* = 0.003). Breeders, who described their activity as the main source of income and the ones labelled as second source of income, played music as a method to de-stress bitches more often than sporadic activity (0.007 and 0.0018, respectively). A statistical effect of the country was observed in relation to the use of natural products (pheromone, Bach flowers, homeopathy, etc.,). It was poorly used in countries like Poland and Portugal (0.0% and 4.9%, respectively) to up to 21.6% in Germany. The use of natural products to reduce stress after parturition was significantly higher for breeders from Germany than from Brazil (*p* = 0.03), Portugal (*p* = 0.02), United Kingdom (*p* = 0.02) and other countries (*p* = 0.005). The use of natural products was equally distributed amongst users, except for oral calming supplementation and phytotherapy (Figure 17).

Overall, 50.9% of the breeders—all countries pooled together—stated that they never witnessed inappropriate maternal behaviour in their bitches. For the other half of them (Table 2), primiparous bitches, in particular, could potentially fail to attach and take care of puppies (98.2%; 322/328). In terms of delivery (Table 3), inappropriate maternal behaviour was mostly observed after a C-section (91.1%; 316/347) while very rarely after normal parturition (8.9%; 31/347). Only 10.3% of responders observed cannibalism by the dam and 31.7% lack of milk production at birth (Table 3).

Puppies stayed together with the dams from 4 weeks to 9 weeks or more based on the country (Figure 18). For modelling, variable ‘length of puppies staying with the mother’ has been classified as 8 weeks or less versus 9 weeks or more, to reflect the most frequent situations observed in the field. Distribution of these lengths was different according to country (*p* < 0.0001). The percentages of dog breeders from Spain, Poland and Portugal keeping puppies with their mother for 9 weeks or more was similar (approximatively 50%), but this was significantly higher than dog breeders from Australia, Germany, Brazil, USA/Canada and other countries.

Overall, breeders from all surveyed countries mostly considered that poor maternal behaviour had an impact on puppies’ cognitive development (83.7%), with some variability: German breeders were the most convinced (98.4%) of this relationship while only 69.7% and 75.8% of Brazilian and Spanish breeders agreed.

## 4. Discussion

The interest of the scientific community has increased over the past five years in relation to the source of puppies, their interaction with the dam and littermates as well as their development and welfare. By trying to understand this synergism, as it may impact the behaviour of the future pet with his new family, more attention is given to gather information about the profile, the goals of dog breeders and the impact of the relationships between veterinarians and dog breeders. However, there is still a lack of key information such as the number of puppies per capita and the core source of puppies for the general population. For instance, it is not only important to better understand breeders and their practices, since their products (puppies) will become the veterinarians’ clients [43], but also because a better communication between veterinarians and breeders can improve the quality and the welfare of dogs. The driving forces for breeding dogs seem to be variable and related to personal choice, lifestyle and/or a simple passion. Nevertheless, breeders can directly affect the health and welfare of dogs due to breeding choices and morphological standards [44,45]. The current study reveals the profile of dog breeders from different countries (Australia, Brazil, Germany, Poland, Portugal, Spain and USA/Canada) as well as their practices and observations in relation to female breeding stock, reproduction, maternal behaviour and stress management.

The characteristics of our sampling and results were comparable to similar studies in Australia [24,26] and France [25,46] where small breeders seem to be overrepresented. Indeed, in relation to their activity, the majority of breeders (90%) considered themselves as small hobby breeders. In general, the responders bred one single breed (79.5%) and they owned less than five reproductive bitches. An equivalent profile has been described in other studies [24,25,26,46]. In Brazil and Spain, larger kennels (>5 bitches) were represented (57.9% and 42.4% respectively). The overrepresentation of large kennels in these two countries is most likely totally random and could be explained by the simple fact that owners of large kennels took the time to answer the survey more than those from other countries. There is no information that supports the fact that in Brazil and/or Spain the number of large kennels is higher than in the other countries here represented. Nevertheless, in Brazil, there is still no federal law that prohibits or regulates the sale of pets in pet shops, nor the commercial breeding of pets what could favour the presence of large kennels. Specific government laws were introduced in the past couple of years in the other represented countries and perhaps a tight control could affect the size of dog breeding facilities. Contradictorily, although in recent years there was an attempt to control puppy farms; dog breeding in the USA was qualified in 2017 as a big business with little governmental oversight [47] and no dog breeders described in this study the activity as their main source of income. In terms of breeds, the representation of dog groups in our sample was in accordance with the popularity of different breeds worldwide.

The housing system varies amongst countries (Figure 4) but for over 95% of responders a close interaction of the dogs with humans was a common practice. A significant difference was observed based on the country; human interactions based on the view of breeders of Portugal were less when compared to USA/Canada (*p* = 0.0299), Germany (*p* = 0.0019), the UK (*p* = 0.0028) and Poland (*p* = 0.0251) and the same trend for breeders of Australia in contrast to Germany (*p* = 0.0091) and the UK (*p* = 0.0144). Breeders from Germany had closer interaction with their dogs than the ones from Spain (*p* = 0.0331). The differences observed could reflect the culture of the relationship with pets in each specific country, but it is hard to pinpoint a single cause. The housing system could partially explain the differences observed: for more than 95% of German responders, the bitches were housed at home or the home and garden and therefore being in contact with the animals was indubitable. On the other hand, when bitches are kept in an outdoor kennel, human interaction might be at specific times such as feeding and cleaning. Interestingly, breeders with five or less bitches spent more time with the animals than breeders with more than five bitches (*p* = 0.0142). This finding is comparable to data from France where, brood bitches are housed in close proximity to the family and the same effect of kennel size was observed [25].

For the reproductive activity, measurement of blood progesterone levels (P4) was one of the most common methods to define the breeding time for over 70% of responders except in the UK (59.5%) and Brazil (50%). Some of the responses might not translate the reality of the practice. Responders may have answered that they frequently used progesterone to give themselves a better status in relation to their expertise, the same is applied to the question of pregnancy diagnosis. However, as stated before, the questionnaire was anonymous and at this point, no clear reason for a deceitful answer was identified, so we believe the answers given truly represented the way of doing of the responders. The use of progesterone levels to define the time of ovulation was also the most common method in France [25]. The second most used technique (more than 50% of responders) was the observation of the bitch’s behaviour during oestrus, with the exception of Brazil (where vaginal smears were used as often as progesterone testing) and Portugal where the use of a teaser male was placed in the second most common technique. The cost/benefit of each technique in relation to the financial return of the selling price of puppies and the economic situation of each country definitely has an impact on the choice of the methods. Unfortunately, the financial status of puppy production by country is not easy to find and highly variable, depending on the breed, the prestige of the breeders, the economic situation of a specific country and puppy’s availability in a capitalist society. Therefore, it is out of the scope of this article to evaluate the possible weight of these factors on the answers of dog breeders. Nevertheless, small hobby breeders (≤5 bitches) will invest and use more expensive methods like progesterone (*p* = 0.008) than larger structures, highlighting the economic impact on their choices. Moreover, the cost of progesterone assays is higher than the other methods listed, but it is the most accurate way to better define the timing and range of the fertile period during oestrus in the bitch [48,49]. Observation of the bitch’s behaviour during oestrus is a simple and practical technique. The limitation is the differences between and within bitches in relation to receptivity and responses to the male dog. Bitch behaviour is not an ideal option to consistently predict the fertile time [50]. Vaginal cytology as a single method is not reliable to detect ovulation prospectively and consequently the optimal time to breed [51]. Although vaginal electrical impedance has been the subject of different studies to evaluate its efficacy in predicting breeding time [52,53], this method was barely listed by breeders. However, vaginal impedometry is faster and cheaper than progesterone assessment and had showed to be more reliable than vaginal cytology and clinical evaluation [52] but was considered as unreliable as a method for monitoring periovulatory events [53] and therefore breeding time. The method used to predict breeding time is very important since the accuracy of the method will influence the pregnancy rate, and mistimed breeding is the most common cause of infertility in the bitch. The importance of pregnancy data in the breeder’s activity was demonstrated by the massive use of ultrasound (for over 80% of responders) to perform pregnancy diagnosis. The differences observed in the methods of choice to confirm pregnancy can be once again related to the cost of the technique for each specific country. Breeders with ≤5 bitches seem to invest more to know if their bitches are pregnant by using abdominal ultrasound as the method of choice compared to breeders with more than 5 bitches (*p* = 0.04)

The estimation of the whelping date as observed in the previous study in France [25] employed a large variation of methods with a large diversity amongst countries. Four methods were equally cited (by around 20% of responders) to estimate the time of parturition (numbers of days from the last day of breeding, bitch’s behavioural changes, body temperature measurement at the end of gestation and numbers of days from day of ovulation). As a group, the dog breeders utilised the methods described in the literature [54], but the country had a significant effect on the choice of each specific method. Financial reasons could explain the choices of dog breeders from different countries, but as stated before these particularities are out of the scope of this article. Nevertheless, when ovulation date was used to predict the breeding time, most likely the same date was used to estimate parturition by breeders from USA/Canada and Australia, since the duration of gestation based on serum levels of progesterone is 65 ± 1 day (LH peak) and 63 ± 1 day (ovulation time). Despite the fact that the sharp drop of progesterone at the end of gestation is a reliable method to predict impending parturition [55], it is not practical and was not described as a method used by this sample of dog breeders. However, it was applied indirectly when breeders cited the measurement of temperature of the bitch at the end of pregnancy since this corresponds with the drop of progesterone. Although, the decrease of rectal temperature may indicate the onset of parturition [54], as a single method it is not very reliable. Measuring the temperature two or three times per day seems to improve the prediction of the whelping date. Yet, the method has a low positive predictive value [56].

The surveillance of the parturition was a common practice amongst breeders from different countries, whatever the number of bitches (≤5 or >5), with over 90% of responders staying close to the bitch or visiting the room frequently when labour was in progress. Devices such as cameras or audio surveillance were surprisingly not described as the main methods applied (10/668; 1.5%), in contrast with almost 10% of French dog’ breeders [25] who relied only on electronic monitoring (video or acoustic) during parturition. The close surveillance of the bitch at whelping time is an important aspect to consider since parturition was stated to be a stressful situation for bitches by over 80% of responders. Various signs were described as signs of stress although some can be associated to normal parturition such as pacing, trembling and restlessness. No common sign was described either as an indicator of over-stressed bitches nor as a specific behaviour of a calm animal. Breeders diligently watched the whelping. Interventions to reduce stress were cited as preventive methods by 68.6% of breeders (457/668) and as a ‘curative’ action (only for bitches who seemed stressed) by 24.2% (161/668). Human presence and increasing the close contact with the bitch were primarily routine methods to manage stress regardless of the country and the size of the kennel. Indeed, overstressed animals can have physiological changes able to influence physiological events. Anxiety can increase heart rate and interferes with contractions during parturition in humans [57] but in dogs, although no research on maternal overstress and the impact on labour has been done, the impact of stress and neonate survival has been reviewed [58]. Since domestic dogs are very attached to humans, being close to the bitch during a stressful time might reassure them and reduce the negative effect of overstress during whelping. Human interaction can improve the welfare of dogs in shelters by reducing the levels of anxiety [59]. Other procedures to reduce stress were influenced by the country, with probably a cultural bias, and by the size of the kennel. Breeders from Portugal and USA/Canada described playing music more often than other places and was a more common practice by breeders whose activity was the main source of income. Breeders from Germany were keener to use natural products than other countries. The use of a specific technique to calm the bitch at the time of parturition could reflect the anthropomorphic trend in each country. If people believe that certain procedures can help make them less stressed, they may be more inclined to use the same on their animals. It could be interesting to look into each country’s lifestyle to see the impact on the way to treat and address problems with alternative medicines and cross the data with veterinary practices. The effect of the size of the kennel was similar to that observed by French breeders [25]: large structures are more likely to use music to reduce stress since having someone with the bitch all the time might be a limiting factor. In this case, music seems to be an effective cheap method. Exposure to music—in particular classical music—seems to have a calming influence on dogs in stressful environments such as shelters [60].

Free access to outside seems to be the normal practice amongst different countries, with more controlled exits from the whelping area described by breeders from Brazil and Spain. Since these two countries also represent the samples of large kennel structures, it is possible that the differences presented were more a reflection of the size of the breeding operation.

Breeders evaluated the quality of ‘motherly style’ based on the frequency of two major behaviours, namely, nursing and licking the newborns. In fact, recent studies have shown that mothers who scored high on maternal care were more often in contact and showed high levels of oral behaviour towards their puppies [61,62,63]. Breeders usually observed the role of motherly behaviour by the bitch soon after parturition. Signs of overstress and/or poor maternal care were associated with rejection and frequent displacement of puppies. Half (49.1%) of the responders observed inappropriate maternal behaviour, almost exclusively in primiparous females.

Similarly to the prenatal and parturition stages, the most common method quoted by breeders (all countries together) to control maternal stress during the postpartum period was to increase the human presence for 86.5% of the survey sample, followed by playing music (22.2%). Then using natural products was the third option (10.3%). Recently, the benefits of using dog appeasing pheromone products (ADAPTIL^®^, Ceva Santé Animale, Libourne, France) during the peripartum and neonatal periods have been studied [64], and found that bitches exposed to the pheromone displayed more attention and dedicated more time to their puppies. Such practical databased evidence may modify the choice of methods to comfort bitches around parturition in the coming years.

The results presented here could be a tool for veterinarians in each listed country to better understand a set of dog breeders’ activity, addressing stress management by providing advice about the different options, and possibly develop further studies about puppy production and animal welfare.

## 5. Limitations

The study was based on a survey of convenience sampling and purely qualitative in nature. The representativeness of our sample (*n* = 668) from all responders combined regardless of their country is definitely too small in relation to the whole dog breeder population. Moreover, despite our efforts to reach as many different types and sizes of kennel owners as possible owners in each target country, our sample most likely does not reflect the general population. Consequently, the results may not apply to the overall population of dog breeders. The survey design might have selected more technological breeders giving then a false interpretation of some specific results, for example the use of progesterone levels and pregnancy diagnosis as a common practice. With the exception of Spanish and Brazilian breeders, the representation of large commercial breeding operations was not in the pool sample. Most likely, the practice, needs, expectations and view of large commercial kennel are probably distinct (one of the reasons of the dissimilarity observed in relation to the answers from Brazil and Spain) and differ from small, show, and hobby breeders. Nevertheless, the results are similar to those previously observed by the same authors in a survey carried out in France. Therefore, we strongly believe that the results of this survey might represent an important segment of puppy production activity, since the responders might be, in some respects, key opinion leaders. Consequently, better knowledge and communication with this group could have a large impact on the whole dog breeding activity. Another aspect is that 20.5% of responders bred several breeds and their answers to most of the questions could not take in consideration each specific breed (note: for questions related to dogs’ behaviour, if respondents bred several dog breeds, they were asked to consider the main breed of their activity). Therefore, the analysis of a possible ‘breed effect’ on the surveyed parameters was not statistically feasible. No cluster of breeds (brachycephalic) more predisposed to require a caesarean section were described in the sample. Another reflection is the comparison of climate of each country that was not taken into consideration in the statistical analysis. Indeed, the climate could have an effect on the housing systems. Since the housing systems could influence the human interaction, these two factors are most likely affected by the weather. Nevertheless, for the other differences or similarities observed in the survey, economic and cultural factors presumably are more important than the climate.

## 6. Conclusions

This work improved our understanding of the rather complex activity of dog breeding and diverse practices used by breeders to assist in the reproductive process and manage peripartum overstress and maternal care of offspring. The recognition that breeders are concerned about the stress of bitches around whelping time (peripartum period) allows the design of strategies to better manage these situations.

The statement that maternal care can be disturbed due to stress and consequently have a direct impact on the development of puppies gives an opportunity for veterinarians to further play a role in the breeding activity and indirectly influence the animals’ welfare.

The more we gather information about dog breeders’ opinions, aims and goals, the more this will facilitate interventions to address reproductive and other issues. In addition, this knowledge can facilitate the communication amongst different players involved directly or indirectly in the dog-breeding activity. Better collaboration and cooperation between veterinarians and dog breeders will undoubtedly have a positive impact on the health of the dam and puppies and therefore on animal welfare.

## Figures and Tables

**Figure 1 animals-11-02372-f001:**
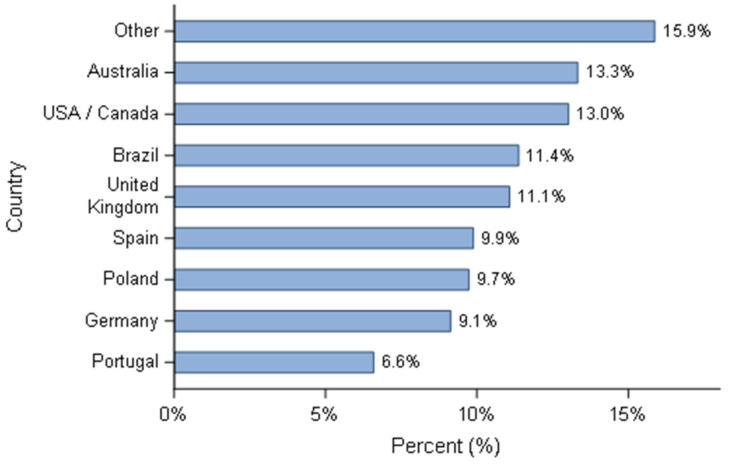
Responders’ distribution according to the country—all responders (*n* = 668).

**Figure 2 animals-11-02372-f002:**
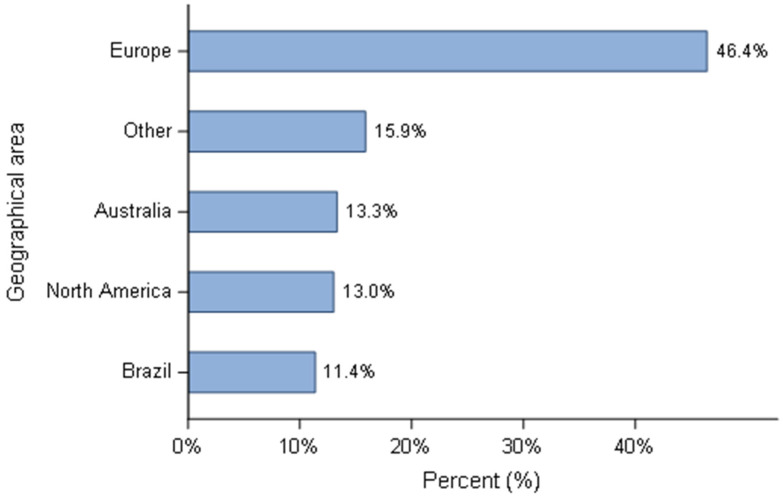
Responders’ distribution grouped by geographical region—all responders (*n* = 668).

**Figure 3 animals-11-02372-f003:**
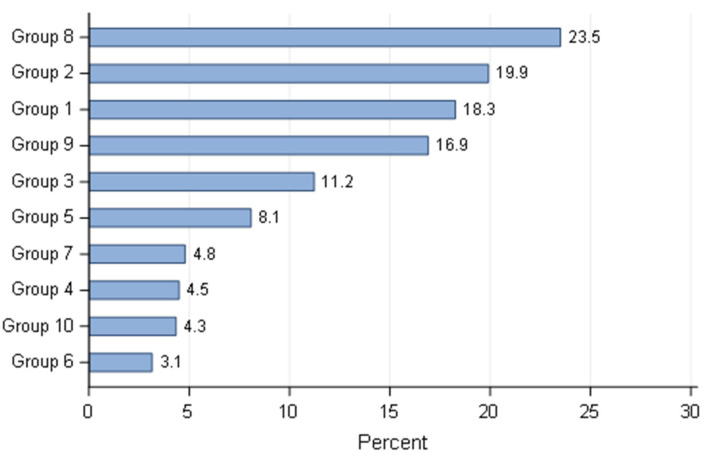
Overall distribution based on different groups according to the classification of the Federation Cynologique Internationale (FCI—www.fci.be (accessed on 4 June 2021) (multiple answers were possible)—all responders (*n* = 668).

**Figure 4 animals-11-02372-f004:**
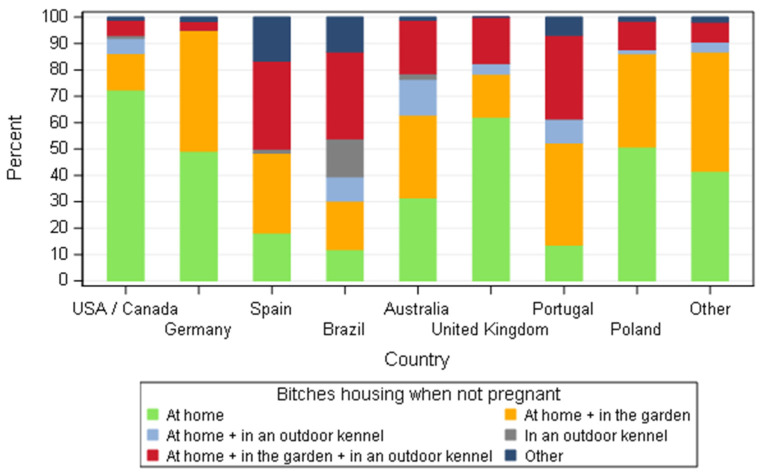
Housing system distribution for non-pregnant bitches according to dog breeders by country—all responders (*n* = 668).

**Figure 5 animals-11-02372-f005:**
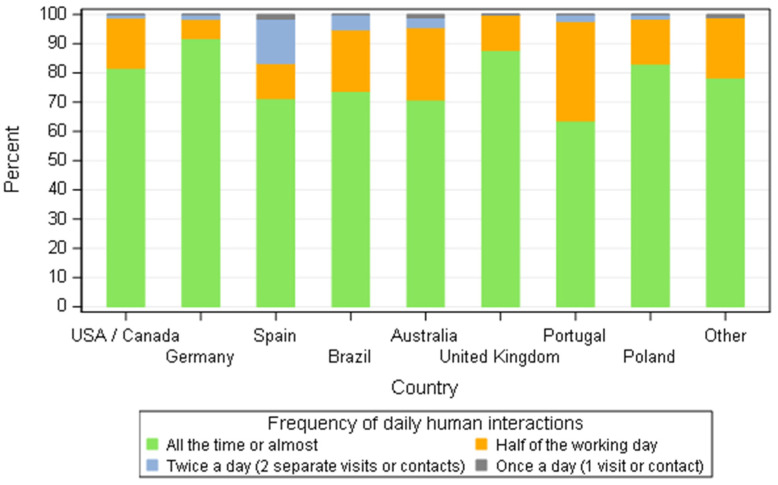
Frequency of daily human interactions distribution according to dog breeders by country—all responders (*n* = 668).

**Figure 6 animals-11-02372-f006:**
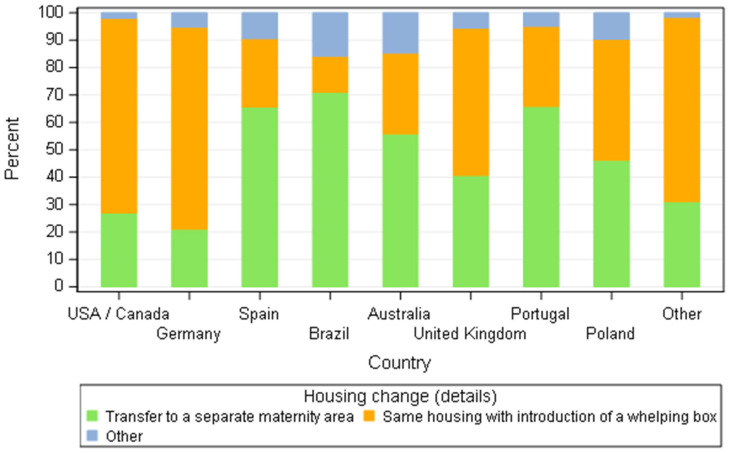
Distribution of bitch housing change around parturition according to dog breeders by country—all responders stating to change the house before parturition (*n* = 488).

**Figure 7 animals-11-02372-f007:**
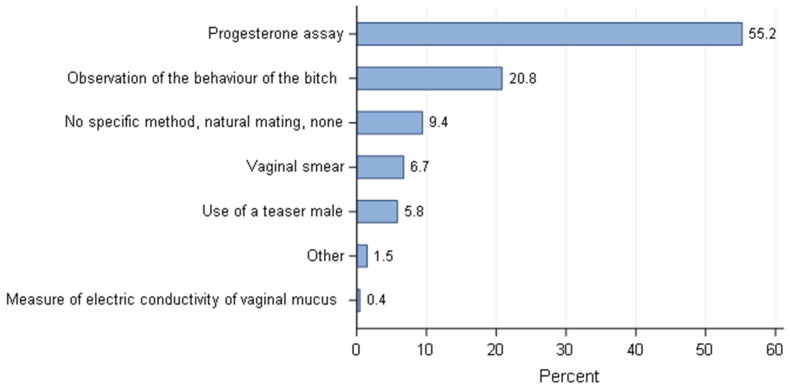
Methods to estimate the breeding time (most often used) overall distribution—all responders (*n* = 668).

**Figure 8 animals-11-02372-f008:**
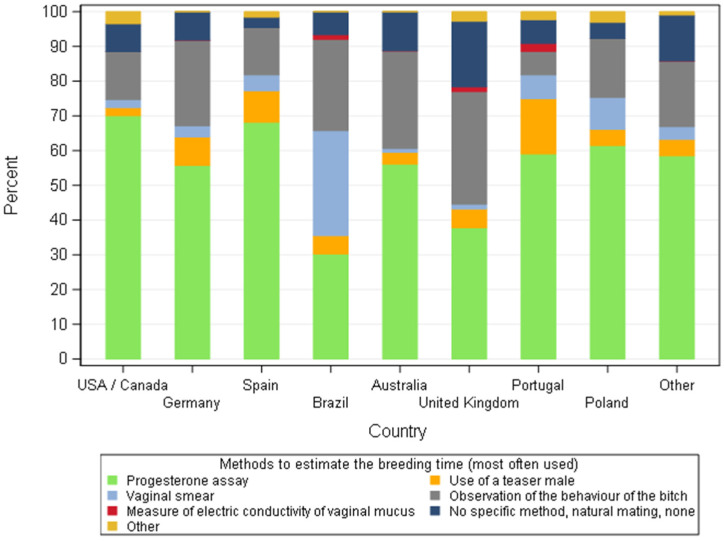
Methods, most often used, to estimate the breeding time according to dog breeders by country—all responders (*n* = 668).

**Figure 9 animals-11-02372-f009:**
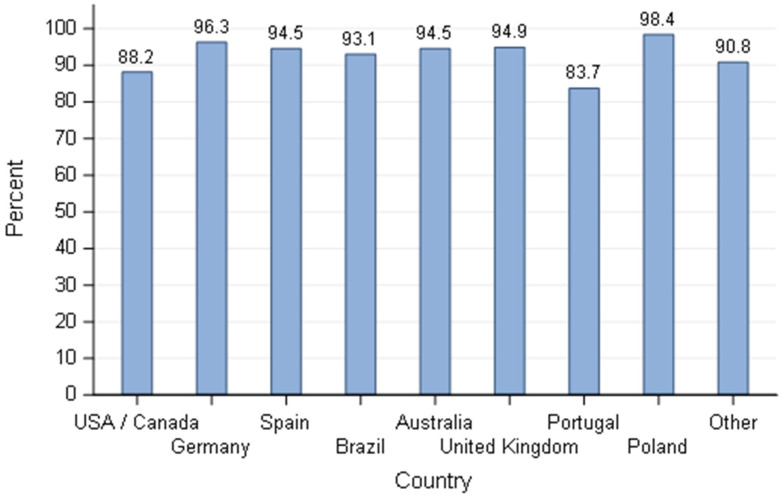
Use of ultrasound as a method to confirm pregnancy in breeding bitches based on breeders’ responses by country—responders stating to do a pregnancy diagnosis (*n* = 585).

**Figure 10 animals-11-02372-f010:**
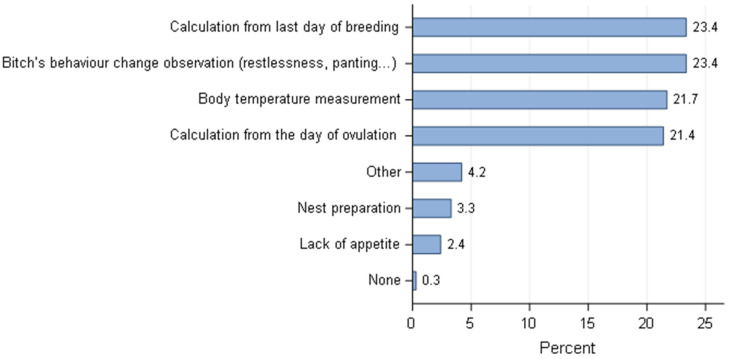
Overall distribution of methods to estimate the whelping time—all responders (*n* = 668).

**Figure 11 animals-11-02372-f011:**
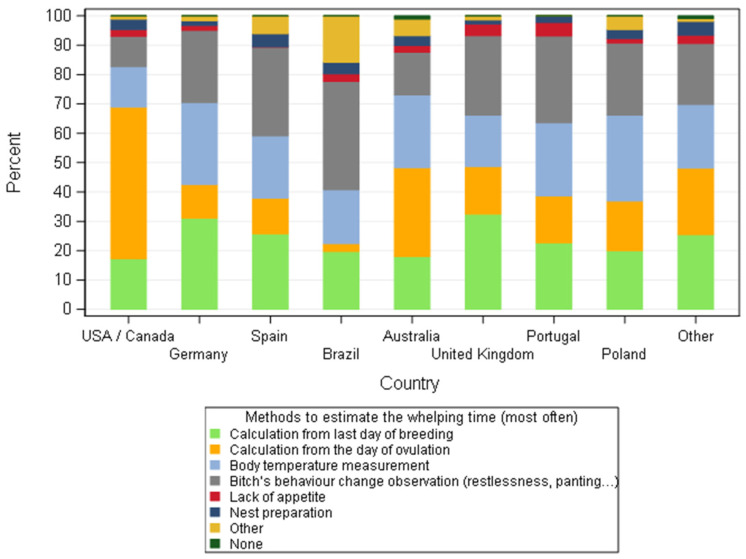
Methods, most often used, to estimate the whelping time according to dog breeders by country—all responders (*n* = 668).

**Figure 12 animals-11-02372-f012:**
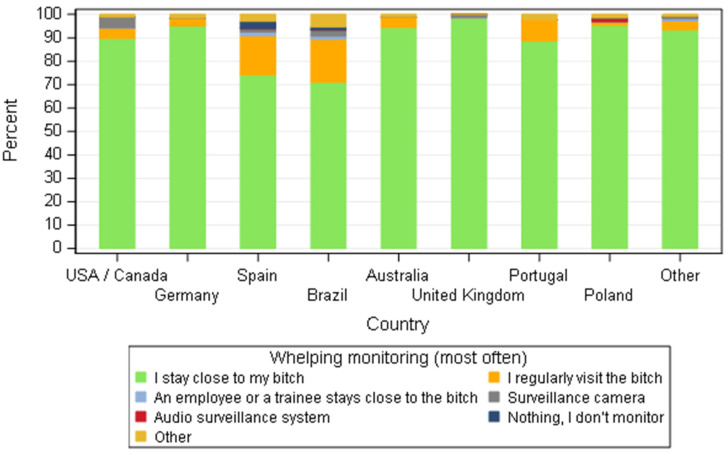
Distribution of the whelping-monitoring system according to dog breeders by country—all responders (*n* = 668).

**Figure 13 animals-11-02372-f013:**
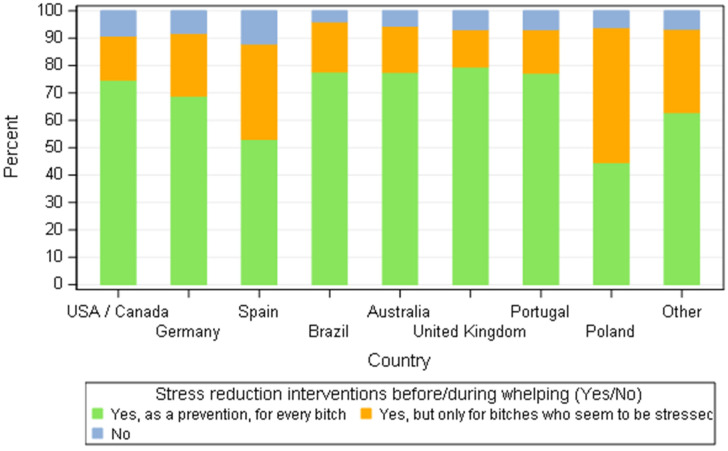
Distribution of responders that intervene to reduce stress before/during whelping by country—all responders (*n* = 668).

**Figure 14 animals-11-02372-f014:**
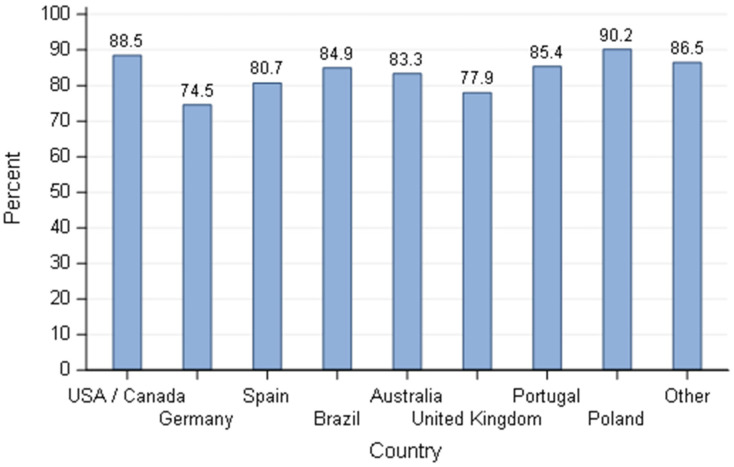
Percentage of dog breeders in the survey that spend more time with the bitch as an intervention to reduce stress before/during whelping by country—responders stating to intervene (*n* = 618).

**Figure 15 animals-11-02372-f015:**
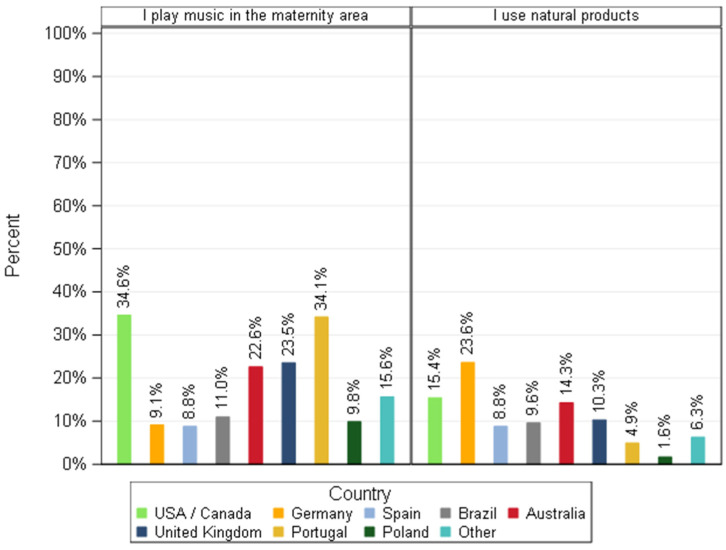
Percentage of dog breeders that use music and natural products as an intervention to reduce stress before/during whelping by country—responders stating to intervene (*n* = 618).

**Figure 16 animals-11-02372-f016:**
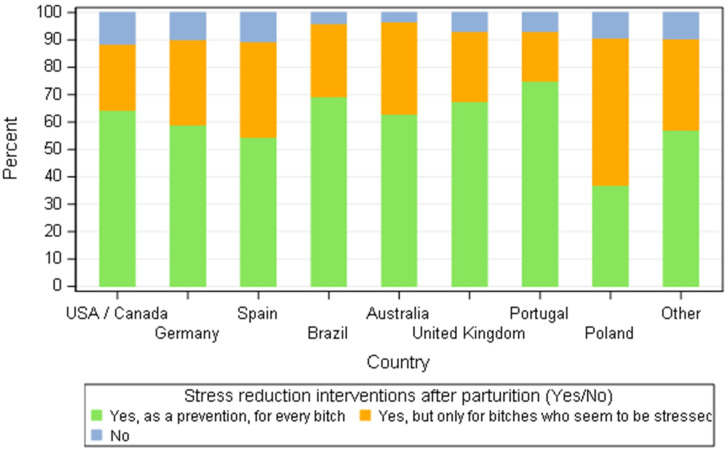
Distribution of responders that intervene to reduce stress after parturition by country—all responders (*n* = 668).

**Figure 17 animals-11-02372-f017:**
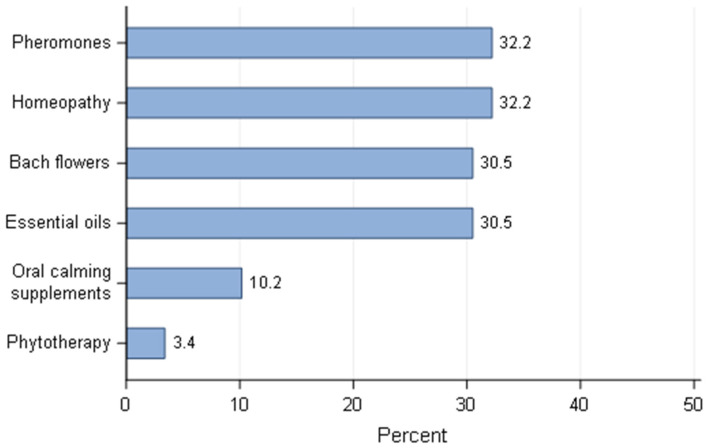
Distribution of the use of natural products to reduce stress (after parturition)—responders stating to use natural products (*n* = 62).

**Figure 18 animals-11-02372-f018:**
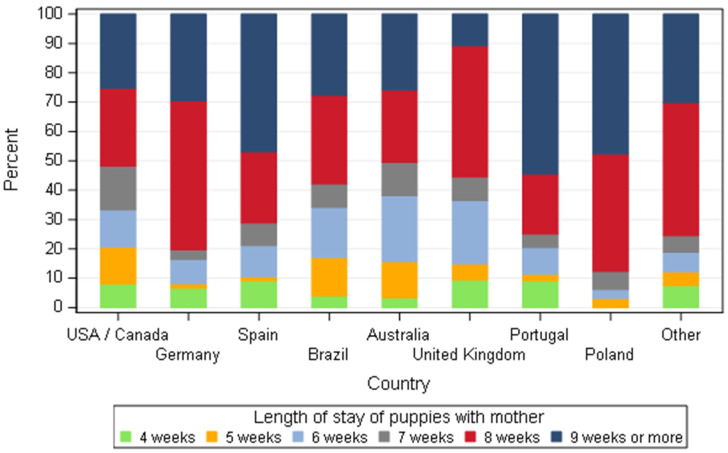
Duration in weeks that puppies stay with the dam according to dog breeders by country—all responders (*n* = 668).

**Table 1 animals-11-02372-t001:** Survey structure and main content.

Section 1: General information	Contact details of the breeder (country, type of activity—professional or familiar) Number of breeding bitches and litters per year Dog types and specific breeds Animal housing systems Level of interaction with humans.
Section 2: Breeding practices	Specific techniques to manage reproduction Changes in housing system and exercise schedule of the bitch around parturition.
Section 3: Parturition	Timing parturition Whelping monitoring Signs of maternal stress Peripartum stress management.
Section 4: After parturition	Exercise schedule Maternal behaviour in peripartum Signs of poor maternal care Methods used to reduce the stress and improve maternal care Observation of inappropriate maternal behaviour.
Section 5: Generalities	General thoughts about maternal behaviour Time that the dam and puppies stay together Potential behaviour differences among breeds.

**Table 2 animals-11-02372-t002:** Inappropriate maternal behaviour according to parity based on different countries (*n* = 668).

Parameter	Statistics	Country									
		USA and Canada	Germany	Spain	Brazil	Australia	United Kingdom	Portugal	Poland	Other	Total
		(*n* = 87)	(*n* = 61)	(*n* = 66)	(*n* = 76)	(*n* = 89)	(*n* = 74)	(*n* = 44)	(*n* = 65)	(*n* = 106)	(*n* = 668)
Primiparous bitches (first whelping)	% (*n*)	50.6% (44)	24.6% (15)	63.6% (42)	72.4% (55)	41.6% (37)	39.2% (29)	50.0% (22)	53.8% (35)	40.6% (43)	48.2% (322)
Multiparous bitches (experienced bitches)	% (*n*)	1.1% (1)	0.0% (0)	1.5% (1)	2.6% (2)	1.1% (1)	1.4% (1)	0.0% (0)	0.0% (0)	0.0% (0)	0.9% (6)
Not applicable, I never had such problem	% (*n*)	48.3% (42)	75.4% (46)	34.8% (23)	25.0% (19)	57.3% (51)	59.5% (44)	50.0% (22)	46.2% (30)	59.4% (63)	50.9% (340)

**Table 3 animals-11-02372-t003:** Inappropriate maternal behaviour according to type of parturition, puppy cannibalism and lack of presence of milk based on different countries (n = 668).

Parameter	Statistics	Country									
		USA and Canada	Germany	Spain	Brazil	Australia	United Kingdom	Portugal	Poland	Other	Total
		(*n* = 87)	(*n* = 61)	(*n* = 66)	(*n* = 76)	(*n* = 89)	(*n* = 74)	(*n* = 44)	(*n* = 65)	(*n* = 106)	(*n* = 668)
Type of parturition											
Normal parturition	*% (n)*	3.4% (3)	4.9% (3)	6.1% (4)	5.3% (4)	1.1% (1)	4.1% (3)	4.5% (2)	6.2% (4)	6.6% (7)	4.6% (31)
C-section (surgery)	% (*n*)	54.0% (47)	32.8% (20)	53.0% (35)	75.0% (57)	52.8% (47)	33.8% (25)	43.2% (19)	40.0% (26)	37.7% (40)	47.3% (316)
Not applicable, I never had such problem	% (*n*)	42.5% (37)	62.3% (38)	40.9% (27)	19.7% (15)	46.1% (41)	62.2% (46)	52.3% (23)	53.8% (35)	55.7% (59)	48.1% (321)
**Puppy cannibalism from bitches**											
Yes	% (*n*)	11.5% (10)	1.6% (1)	19.7% (13)	21.1% (16)	11.2% (10)	8.1% (6)	9.1% (4)	3.1% (2)	6.6% (7)	10.3% (69)
No	% (*n*)	88.5% (77)	98.4% (60)	80.3% (53)	78.9% (60)	88.8% (79)	91.9% (68)	90.9% (40)	96.9% (63)	93.4% (99)	89.7% (599)
**Lack of milk**											
Yes	% (*n*)	33.3% (29)	24.6% (15)	47.0% (31)	46.1% (35)	36.0% (32)	24.3% (18)	20.5% (9)	21.5% (14)	27.4% (29)	31.7% (212)
No	% (*n*)	66.7% (58)	75.4% (46)	53.0% (35)	53.9% (41)	64.0% (57)	75.7% (56)	79.5% (35)	78.5% (51)	72.6% (77)	68.3% (456)

## Data Availability

Data are available upon reasonable request.

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
