# Peer review of "Profile of Dogs’ Breeders and Their Considerations on Female Reproduction, Maternal Care and the Peripartum Stress—An International Survey"

_animals, 2021, doi:10.3390/ani11082372_

Round 1
Reviewer 1 Report
This MS is to understand the factors considered important in relation to reproduction, maternal behaviour, and the peripartum stress in bitches and demographics of breeders from different countries. The objectives are within the scope of the journal. However, various issues should be fixed:
1-First, the title needs to be modified. Please modify. "the peripartum stress" would be better in the title.
2-Introduction section is very long and not relevant to the main objectives of the study. The authors should present the rationale of the study (why the authors did this study?)
3-Statistical analyses need to be described in details (type of statistical assay, parameters such as nonparametric tests ;chi square, .....). In addition, the significance should be indicated in the figures.
4-Different parameters in the investigation should be indicated as an illustrating chart in the material and methods section.
5-Environmental factors and effect of seasonality were not indicated in this study. The authors should highlight these factors. Otherwise, it should be indicated in the limitations section.
Author Response
Dear Reviewer,
Thank you for the time and effort to review our article. We appreciate your comments and suggestions. We believe that your feedback improved our manuscript. We have incorporated most of the suggestions and only one request was not addressed as described below.
1-First, the title needs to be modified. Please modify. "the peripartum stress" would be better in the title. Done
2-Introduction section is very long and not relevant to the main objectives of the study. The authors should present the rationale of the study (why the authors did this study?) The suggestion was addressed and the introduction was rephrased to better adapt to the objectives of the study.
3-Statistical analyses need to be described in details (type of statistical assay, parameters such as nonparametric tests ;chi square, .....). In addition, the significance should be indicated in the figures. The suggestions were also addressed, however for the p values in the figures; it would not be relevant since the comparison amongst countries was always done two by two.
4-Different parameters in the investigation should be indicated as an illustrating chart in the material and methods section. Done
5-Environmental factors and effect of seasonality were not indicated in this study. The authors should highlight these factors. Otherwise, it should be indicated in the limitations section. Done

Reviewer 2 Report
Interesting article about dog breeders and the influence of the breeding facility on stress and maternal care. It also describes trends in current dog breeding.
I miss the row numbers in the manuscript. It would be much easier to point the exact part of the article.
Page 3, 2nd paragraph- there is a duplication in the sentence after citation 24 "... the shortage of puppies in the UK due to shortage of puppies..."
In figure 1 and 2, page 6 and 7, there are switched captions of the graph axes.
Part 3 Results: as a part of the Results, there is a chapter "4. Management, signs of stress and .....". It should follow the numbering of chapters so it should be "3.4." and the following chapter "4.1 Signs of stress.." should be "3.4.1". Same with the chapter 4.2, it should be "3.4.2". Discussion should be 4, not 5. etc.
I suggest to check the formatting of the text and unify all the fonts etc.
Authors summarize the limitations of this study by themselves very well. I think that it perfectly points out the need of further research in this field to obtain valuable data that could be applied worldwide. At the same time, it also stresses the importance of smaller studies on local scale to find out differences between breeding practices in different countries.
Author Response
Dear Reviewer,
Thank you for the time and effort to review our article. We are grateful for the insightful comments and valuable suggestions. All your corrections were addressed.
I miss the row numbers in the manuscript. It would be much easier to point the exact part of the article. It was addressed.
Page 3, 2nd paragraph- there is a duplication in the sentence after citation 24 "... the shortage of puppies in the UK due to shortage of puppies..." It was addressed.
In figure 1 and 2, page 6 and 7, there are switched captions of the graph axes. It was addressed.
Part 3 Results: as a part of the Results, there is a chapter "4. Management, signs of stress and .....". It should follow the numbering of chapters so it should be "3.4." and the following chapter "4.1 Signs of stress.." should be "3.4.1". Same with the chapter 4.2, it should be "3.4.2". Discussion should be 4, not 5. etc. It was addressed.
I suggest to check the formatting of the text and unify all the fonts etc. It was addressed.

Reviewer 3 Report
This study describes the usual behaviour and breeding methods of dog breeders, throughout an international survey. The manuscript is interesting and innovative and has a good study design. It might be interesting for the reader, therefore, in my opinion, it needs only minor revisions. English language is good and easily readable, but a check is needed in order to find and correct some errors.
It was difficult to review this manuscript because of the lack of line numbering. I try to list my revisions dividing them in sections and when the section was too long in paragraphs, according to the divisions made by the authors in the manuscript.
- Introduction: the manuscript is interesting and easily readable, but it is really long. Please try to summarize the introduction section.
- Please check figures and tables for layout, correspondence with caption and readability of all the words included.
Simple summary
- Line 1: “more and more and more”, please modify in “an increasing number of”
- Lines 8-9: please modify in “The survey focused on breeders’ demographics and maternal behaviour and stress during the peripartum period”.
- Line 11: In this section authors say that “puppy production is an activity driven partially by small-scale breeders”, whereas in the result section they report that for more than the 90% of responders consider breeding activity as a hobby or sporadic activity. Please modify the sentence in the summary section.
- Line 14: Please delete “However,”
Abstract
- Line 2: please modify “member” in “members”
- Lines 3-4: please rephrase this sentence.
- Line 11: please modify “to seize” in “to understand”
- Line 12: please delete the dash between dam and during
- Line 15: please modify “dog’s livelihood” in “the livelihood of the dog”
- Lines 16-17: later in the text authors talk about countries called “others” in which the survey was sent, they are all listed here? If not please add them or “others”.
- Line 23: please correct “was” in “were”
Introduction:
Paragraph 1:
- Please modify “dog’s strategies of adaptation” in “strategies of adaptation of dogs”, check it throughout the manuscript
Paragraph 2:
- Please rephrase the first sentence, the connection between the first and the second part of the sentence is not clear and “although” seems not to be the right conjunction to be used here
- Please modify “dog’s welfare” in the welfare of dogs”
Paragraph 3:
- Please rephrase this sentence: “A recent publication discussed … during the COVID-19 pandemic” avoiding the double repetition of “due to”.
- Please modify “in recent year” in “in recent years”
Paragraph 4:
- Please modify “looks” and the following “look” in “appearance”
Paragraph 5:
- Please modify “behaviourally” in “behaviour”
- Please modify “relinquishment” in “abandonment”
Methods
Section 2.1
Paragraph 1:
- Please modify “We collected data via” in “Data were collected by the use of”
- “(AF, AB, CM, and NS)” this information seems to be unnecessary please delete
Paragraph 2:
- First sentence “The questionnaire, an updated version of a survey conducted in France in 2019,”: please add a citation
- If possible, could the authors insert here who the project’s sponsor is?
Paragraph 3:
- Please modify the first sentence as “The different question used were listed”. In this table all the question used are listed or only the survey structure and the main content are reported? Please check the table caption and content
- The questions used were multiple-choice or open questions?
Table 1:
- Is this the complete survey used or just the different sections of the survey? If the second hypothesis is correct how the different sections were structured? Please consider adding the whole survey in the manuscript.
- In the first line, please modify “family” in “familiar”
- Please in the second column use a different line for each sentence included in each section. Please pay attention to the punctuation used within the table.
Section 2.2
- Please modify “<=” in “≤”
- “For relevant parameters, a logistic…” and “For some parameters, other factors…” please specify what parameters
- Lines 5-6: please rephrase
Section 2.3
- Lines 6-9: Please could the authors better explain what they mean in this sentence. Particularly what do they mean with “portability”?
Results
Section 3.1
- “other categories”, please could the authors list what were the countries included in this category?
- “the profiles of the kennels agreed with the definition of the activity”, please could the authors explain better this sentence?
- “produced more than 11 liters per year”, please correct “liters” in “litters”. How many litters per year have the other countries?
- 3: Please add (www.fci.be) and (n=…) for the number of surveys used
Section 3.2
- Fig 5: Please could the authors explain what they mean with “half of the working day”?
- Fig 6: it seems that here there is an error, because fig 6 graph is the same of fig 5. Please correct.
Section 3.3
- Lines 8-10: please rephrase this sentence
- Fig 7-8: please could the authors explain what is the difference between “no specific method, natural mating” and “none”?
- Fig 9: please could the authors indicate the countries in the x-axis as for the other figures instead of using different colours?
- “to appraise” please modify in “to predict”
- “For European breeders, the distribution of different methods was somewhat similar”, please rephrase this sentence.
- (figure 11), please check the author’s guidelines and use or (Figure x) or (Fig. x) throughout the manuscript
- “scratching”, did the authors mean to dig?
Section 4
- Fig 12: Please control that all words of the legend box could be readable (e.g. bitch)
Section 4.1
- Lines 7-13: please rephrase these sentences
- 14: please could the authors indicate the countries in the x-axis as for the other figures instead of using different colours?
- “the use natural products and providing massage to the dam”: please modify in “providing massage to the dam and the use of natural products”
- Figure 15: please could the authors add information about “massage to the dam” also?
- “the least often considered options”, please modify in “the less often considered options”
Section 4.2
- Lines 13-14: please rephrase
- Line 16: Please modify “Right after..” in “right after..”
- Line 22: please modify “spending more time with the dam” in “spending more time with her”
- Line 35: please modify “It was neglected” in “It was not or poorly used”
- Line 39: please add the p-value (P=……)
- Line 40: looking at Fig 17 it seems that also phytotherapy was less common compared to the other natural products, please add it in the text.
- “The duration that puppies stayed with…” please rephrase
- “Length of stay of puppies” please modify removing capital letter and add “with mother”
- “Reclassified” please modify in “classified”
- “some variability however:” please delete “however”
Discussion
Paragraph 1
- Line 3: Please modify “as well as the development and welfare of puppies” in “as well as their development and welfare”
- Please delete (puppies)
Paragraph 2
- “This overrepresentation of large kennels in these two countries is most likely to be totally random and could be explained by the simple fact that owners of large kennels took the time to answer the survey more than those from other countries. Nevertheless, in Brazil, there is still no federal law that prohibits or regulates the sale of pets in pet shops, nor the commercial breeding of pets what could favor the presence of large kennels. Specific government laws were introduced in the last couple of years in the other represented countries and perhaps a tight control could affect the size of dog breeding facilities. Con-tradictorily, although in recent years there was an attempt to control puppy farms; dog breeding in the USA was qualified in 2017 as a big business with little governmental oversight [49] and no dog breeders described in this study the activity as their main source of income. In terms of breeds, the representation of dog groups in in our sample was in accordance with the popularity of different breeds worldwide.” This part seems to be not really clear, please could the authors try to clearly explain why this finding of this overrepresentation of large kennels in these two countries could be totally random?
Paragraph 3
- “breeders of Portugal declared lower levels of interaction compared to USA/Canada (p=0.0299), Germany (p=0.0019), the UK (p=0.0028) and Poland (p=0.0251) and from breeders of Australia in contrast to Germany (p=0.0091) and the UK (p=0.0144).” please rephrase
Paragraph 4
- “Portugal (use of a teaser male)”, use of teaser male was the most common technique used in Portugal?
- “(p=0.008)” please delete if already mentioned in the result section. Please check it throughout the discussion section.
- “Although vaginal electrical impedance has been the subject of different studies to evaluate its efficacy in predicting breeding time, this method was barely listed by breeders.” Please add citations to this sentence.
Paragraph 5
- “However, it was applied indirectly when breeders cited the measurement of temperature of the bitch at the end of pregnancy since this corresponds with the drop of progesterone.” Please explain what is the thought of the authors about he use of rectal temperature instead of serum progesterone, taking into account the sensibility and specificity of these two different parameters.
Paragraph 6
- “is reviewed” please modify in “has been reviewed”
Paragraph 7
- Lines 3-4: please rephrase this sentence.
Paragraph 9
- “breeders (all countries together)” please modify in “all the interviewed breeders”
Limitations
- “savvy” is too informal, please change with a synonym
- “Therefore, we strongly believe the results..” please modify in “Therefore, we strongly believe that the results..”
- “Key Opinion Leaders”, why the authors used capital letters?
- “specific breed”: this may influence for example answers about the prediction of parturition day and pregnancy monitoring if the breed is a brachycephalic one and if it requires a planned C-section. If possible, consider also this point.
Conclusion
- “around whelping time (prior and after)” please consider to modify in “in the peripartum period”
- “the more it will” please modify in “the more this will”
- “and therefore animal welfare” please modify in “and therefore on animal welfare”
Author Response
Dear Reviewer,
Thank you for the time and effort to review our article. Your insightful and valuable suggestions and comments contribute to improve the quality of our manuscript. The article was re-red by a native English speaker before and after the corrections were done.
We are sorry for the lack of the lines to make it easier for the review and the problem has now been fixed. Once again thank you for your effort to make it easier for us to address the correction. All suggestions and corrections were taken into consideration.
Introduction: the manuscript is interesting and easily readable, but it is really long. Please try to summarize the introduction section. Done
- Please check figures and tables for layout, correspondence with caption and readability of all the words included. Done
Simple summary
- Line 1: “more and more and more”, please modify in “an increasing number of” Done
- Lines 8-9: please modify in “The survey focused on breeders’ demographics and maternal behaviour and stress during the peripartum period”. Done
- Line 11: In this section authors say that “puppy production is an activity driven partially by small-scale breeders”, whereas in the result section they report that for more than the 90% of responders consider breeding activity as a hobby or sporadic activity. Please modify the sentence in the summary section. Done
- Line 14: Please delete “However,” Done
Abstract
- Line 2: please modify “member” in “members” Done
- Lines 3-4: please rephrase this sentence. Done
- Line 11: please modify “to seize” in “to understand” Done
- Line 12: please delete the dash between dam and during Done
- Line 15: please modify “dog’s livelihood” in “the livelihood of the dog” Done
- Lines 16-17: later in the text authors talk about countries called “others” in which the survey was sent, they are all listed here? If not please add them or “others”. Done
- Line 23: please correct “was” in “were” Done
Introduction: The introduction was reduced in length to respond to some of the suggestions of the reviewers. All corrections were taken into consideration.
Paragraph 1:
- Please modify “dog’s strategies of adaptation” in “strategies of adaptation of dogs”, check it throughout the manuscript Done
Paragraph 2:
- Please rephrase the first sentence, the connection between the first and the second part of the sentence is not clear and “although” seems not to be the right conjunction to be used here Done
- Please modify “dog’s welfare” in the welfare of dogs” Done
Paragraph 3:
- Please rephrase this sentence: “A recent publication discussed … during the COVID-19 pandemic” avoiding the double repetition of “due to”. Done
- Please modify “in recent year” in “in recent years” Done
Paragraph 4:
- Please modify “looks” and the following “look” in “appearance” Done
Paragraph 5:
- Please modify “behaviourally” in “behaviour” Done
- Please modify “relinquishment” in “abandonment” Done
Methods
Section 2.1
Paragraph 1:
- Please modify “We collected data via” in “Data were collected by the use of” Done
- “(AF, AB, CM, and NS)” this information seems to be unnecessary please delete Done
Paragraph 2:
- First sentence “The questionnaire, an updated version of a survey conducted in France in 2019,”: please add a citation Done
- If possible, could the authors insert here who the project’s sponsor is? The sponsor is cited in the funding
Paragraph 3:
- Please modify the first sentence as “The different question used were listed”. In this table all the question used are listed or only the survey structure and the main content are reported? Please check the table caption and content Done
- The questions used were multiple-choice or open questions? Done
Table 1:
- Is this the complete survey used or just the different sections of the survey? If the second hypothesis is correct how the different sections were structured? Please consider adding the whole survey in the manuscript.. There is a link to the survey at the end of the article
- In the first line, please modify “family” in “familiar” Done
- Please in the second column use a different line for each sentence included in each section. Please pay attention to the punctuation used within the table. Done
Section 2.2
- Please modify “<=” in “≤” Done
- “For relevant parameters, a logistic…” and “For some parameters, other factors…” please specify what parameters Done
- Lines 5-6: please rephrase Done
Section 2.3
- Lines 6-9: Please could the authors better explain what they mean in this sentence. Particularly what do they mean with “portability”?
- Portability is the common term used in GDPR. The data subject shall have the right to receive the personal data concerning him or her, which he or she has provided to a controller, in a structured, commonly used and machine-readable format and have the right to transmit those data to another controller.
Results
Section 3.1
- “other categories”, please could the authors list what were the countries included in this category? It is not possible since we cannot know from which country was the person who selection the “other” option.
- “the profiles of the kennels agreed with the definition of the activity”, please could the authors explain better this sentence? Done
- “produced more than 11 liters per year”, please correct “liters” in “litters”. How many litters per year have the other countries? Added
- 3: Please add (www.fci.be) and (n=…) for the number of surveys used Done
Section 3.2
- Fig 5: Please could the authors explain what they mean with “half of the working day”? Done
- Fig 6: it seems that here there is an error, because fig 6 graph is the same of fig 5. Please correct. Done
Section 3.3
- Lines 8-10: please rephrase this sentence Done
- Fig 7-8: please could the authors explain what is the difference between “no specific method, natural mating” and “none”? There is actually no difference, thus both options have been pooled in the figures.
- Fig 9: please could the authors indicate the countries in the x-axis as for the other figures instead of using different colours? Done
- “to appraise” please modify in “to predict” Done
- “For European breeders, the distribution of different methods was somewhat similar”, please rephrase this sentence. Done
- (figure 11), please check the author’s guidelines and use or (Figure x) or (Fig. x) throughout the manuscript Done
- “scratching”, did the authors mean to dig? It was better explained
Section 4
- Fig 12: Please control that all words of the legend box could be readable (e.g. bitch) The font of the figures has been modified.
Section 4.1
- Lines 7-13: please rephrase these sentences Done
- 14: please could the authors indicate the countries in the x-axis as for the other figures instead of using different colours? Done
- “the use natural products and providing massage to the dam”: please modify in “providing massage to the dam and the use of natural products” Done
- Figure 15: please could the authors add information about “massage to the dam” also? Contrary to playing music and using natural products, where a country effect was observed, providing massage to the bitch was more common and homogenous amongst countries. In our opinion, this figure would not bring added value to the paper.
- “the least often considered options”, please modify in “the less often considered options” Done
Section 4.2
- Lines 13-14: please rephrase Done
- Line 16: Please modify “Right after..” in “right after..” Done
- Line 22: please modify “spending more time with the dam” in “spending more time with her” Done
- Line 35: please modify “It was neglected” in “It was not or poorly used” Done
- Line 39: please add the p-value (P=……) Added
- Line 40: looking at Fig 17 it seems that also phytotherapy was less common compared to the other natural products, please add it in the text. Done
- “The duration that puppies stayed with…” please rephrase Done
- “Length of stay of puppies” please modify removing capital letter and add “with mother” Done
- “Reclassified” please modify in “classified” Done
- “some variability however:” please delete “however” Done
Discussion
Paragraph 1
- Line 3: Please modify “as well as the development and welfare of puppies” in “as well as their development and welfare” Done
- Please delete (puppies) Done
Paragraph 2
- “This overrepresentation of large kennels in these two countries is most likely to be totally random and could be explained by the simple fact that owners of large kennels took the time to answer the survey more than those from other countries. Nevertheless, in Brazil, there is still no federal law that prohibits or regulates the sale of pets in pet shops, nor the commercial breeding of pets what could favor the presence of large kennels. Specific government laws were introduced in the last couple of years in the other represented countries and perhaps a tight control could affect the size of dog breeding facilities. Contradictorily, although in recent years there was an attempt to control puppy farms; dog breeding in the USA was qualified in 2017 as a big business with little governmental oversight [49] and no dog breeders described in this study the activity as their main source of income. In terms of breeds, the representation of dog groups in in our sample was in accordance with the popularity of different breeds worldwide.” This part seems to be not really clear, please could the authors try to clearly explain why this finding of this overrepresentation of large kennels in these two countries could be totally random? Done
Paragraph 3
- “breeders of Portugal declared lower levels of interaction compared to USA/Canada (p=0.0299), Germany (p=0.0019), the UK (p=0.0028) and Poland (p=0.0251) and from breeders of Australia in contrast to Germany (p=0.0091) and the UK (p=0.0144).” please rephrase Done
Paragraph 4
- “Portugal (use of a teaser male)”, use of teaser male was the most common technique used in Portugal? Done
- “(p=0.008)” please delete if already mentioned in the result section. Please check it throughout the discussion section. We opted to leave the p values during the discussion since the article presents large quantity of data and it might make it easier for the reader to see when the differences were important. Unless it is a major error to do at the discussion, we would like to keep the p values.
- “Although vaginal electrical impedance has been the subject of different studies to evaluate its efficacy in predicting breeding time, this method was barely listed by breeders.” Please add citations to this sentence. Done
Paragraph 5
- “However, it was applied indirectly when breeders cited the measurement of temperature of the bitch at the end of pregnancy since this corresponds with the drop of progesterone.” Please explain what is the thought of the authors about the use of rectal temperature instead of serum progesterone, taking into account the sensibility and specificity of these two different parameters. Done
Paragraph 6
- “is reviewed” please modify in “has been reviewed” Done
Paragraph 7
- Lines 3-4: please rephrase this sentence. Done
Paragraph 9
- “breeders (all countries together)” please modify in “all the interviewed breeders” Done
Limitations
- “savvy” is too informal, please change with a synonym Done
- “Therefore, we strongly believe the results..” please modify in “Therefore, we strongly believe that the results..” Done
- “Key Opinion Leaders”, why the authors used capital letters? Done
- “specific breed”: this may influence for example answers about the prediction of parturition day and pregnancy monitoring if the breed is a brachycephalic one and if it requires a planned C-section. If possible, consider also this point. Done
Conclusion
- “around whelping time (prior and after)” please consider to modify in “in the peripartum period” Done
- “the more it will” please modify in “the more this will” Done
- “and therefore animal welfare” please modify in “and therefore on animal welfare” Done

Reviewer 4 Report
The paper presents the results of a survey intended to study the demographics of dog breeders and to seize their perception and practices about reproduction, maternal behavior and management of the dam-during the peripartum period.
The data presented are numerous, some of which are not very interesting (for example the specific techniques used to estimate breeding). It would have been useful to know the cultural level of the breeders and if they have taken courses in dog biology, ethology and genetics. Unfortunately, cases of "genetic maltreatment" with the birth of animals carrying hereditary defects are more and more frequent.
It would also have been useful to try to understand if altered behavior at delivery or the stress signs are related to the breeding methods.
Overall, the paper is interesting for the large number of data it proposes, even if a more critical approach would have been appreciated.
I believe it can be accepted for publication
Author Response
Dear Reviewer,
Thank you for the time and effort to review our article. We appreciate your valuable comments. Indeed, some important issues were not addressed in the survey. We were conscious of this constraint but the survey used was initially run in France and we wanted to have the possibility to compare the results. We reduced the introduction and added some more critical discussion over some points. We hope these modifications improved the current version, in order to have a better chance to publish the article. Once again, thank you for your time and comments.
The paper presents the results of a survey intended to study the demographics of dog breeders and to seize their perception and practices about reproduction, maternal behavior and management of the dam-during the peripartum period.
The data presented are numerous, some of which are not very interesting (for example the specific techniques used to estimate breeding). It would have been useful to know the cultural level of the breeders and if they have taken courses in dog biology, ethology and genetics. Unfortunately, cases of "genetic maltreatment" with the birth of animals carrying hereditary defects are more and more frequent.
It would also have been useful to try to understand if altered behavior at delivery or the stress signs are related to the breeding methods.
Overall, the paper is interesting for the large number of data it proposes, even if a more critical approach would have been appreciated.
I believe it can be accepted for publication

Round 2
Reviewer 1 Report
The authors addressed the comments raised in the round 1. So, I endorse acceptance in the current form.